# Membrane Surface Modification via In Situ Grafting of GO/Pt Nanoparticles for Nitrate Removal with Anti-Biofouling Properties

**DOI:** 10.3390/mi14010128

**Published:** 2023-01-03

**Authors:** Mohammad Khajouei, Mahsa Najafi, Seyed Ahmad Jafari, Mohammad Latifi

**Affiliations:** 1Department of Chemical Engineering, Polytechique Montréal, Montréal, QC H3T 1J4, Canada; 2Department of Chemical Engineering, Vrije Universiteit Brussel, Pleinlaan 2, 1050 Brussels, Belgium; 3Department of Chemical and Process Engineering, University of Bologna, 40126 Bologna, Italy

**Keywords:** nanoparticle, water treatment, ultrafiltration, environmental engineering, nitrate removal, antibacterial, anti-biofouling

## Abstract

Nanofiltration processes for the removal of emerging contaminants such as nitrate are a focus of attention of research works as an efficient technique for providing drinking water for people. Polysulfone (PSF) nanofiltration membranes containing graphene oxide (GO)/Pt (0, 0.25, 0.5, 0.75, 1 wt%) nanoparticles were generated with the phase inversion pathway. The as-synthesized samples were characterized by FTIR, SEM, AFM, and contact angle tests to study the effect of GO/Pt on hydrophilicity and antibacterial characteristics. The results conveyed that insertion of GO/Pt dramatically improved the biofouling resistance of the membranes. Permeation experiments indicated that PSF membrane embracing 0.75 wt% GO/Pt nanoparticles had the highest nitrate flux and rejection ability. The membrane’s configuration was simulated using OPEN-MX simulating software indicating membranes maintaining 0.75 wt% of GO/Pt nanoparticles revealed the highest stability, which is well in accordance with experimental outcomes.

## 1. Introduction

Groundwater pollution by inorganic salts has been the focus of attention since groundwater is one of the main resources of drinking water in the world [1,2]. Due to the ongoing utilization of nitrogenous fertilizers in the agricultural sector and continuing discharge of effluents which introduce nitrate pollution to groundwater, elimination and control of nitrate in drinking water has been studied to produce clean water for human usage [3]. Nitrate, as a highly soluble contaminant in water, is so toxic to human health at very low concentrations that it can lead to a variety of severe disorders and even death [4]. Therefore, with the aim of protecting human and environmental safety, nitrate contamination in drinking water should be kept to safe levels [5]. A lot of research has been carried out on reducing water pollution from sources of energy, especially nitrate ions, including membrane-based processes, ion exchange, and chemical and biological denitrification [6,7]. Membrane processes, in comparison to other abovementioned processes, are the most promising technique due to having multitudinous benefits in terms of operation, environment, economy, and energy [8]. Membrane technology (specially nanofiltration or NF) as a the high-potential process for removal and separation of inorganic substrates, is categorized as reverse osmosis (RO) or ultrafiltration (UF) membranes, with low affinity for removal of monovalent ions and high capability for removal of multivalent ions [9]. Meanwhile, polysulfone (PSF) is a commercially available polymeric material used in NF membranes [10]. Although PSF polymer offers convenient features like significant thermal, mechanical, and chemical stabilities along with an affordable price, it is susceptible to biofouling owing to its hydrophobic nature [11]. Hydrophobic surfaces of PSF membranes interact with proteins and other foulants [12], whereby a bacterial film forms on the membrane surfaces. In the long run, membrane efficiency reduces and can lead to undesirable results, for example, reduction of water permeability, a decrease of separation efficiency, and the detriment of membrane structure [13]. One of the proposed strategies for solving the biofouling problem is physical and chemical pretreatment of feed solution to exclude bacterial existence. Physical treatment such as UV irradiation is expensive and cannot adjust biofouling in the membrane texture, and chemical treatments including oxidizing and non-oxidizing biocides damage the membrane surfaces, reduce their lifetime, and finally increase operating costs. Moreover, due to the rapid proliferation of microorganisms on the surface of membranes, especially when highly needed due to the amount of nutrient in the feed stream, the complete exclusion of biofouling phenomenon cannot be guaranteed. A long-established effective strategy for reduction of biofouling is a donation of cytotoxicity properties to the membrane through surface adjustment via improving the membrane’s hydrophilicity. Hydrophilic membranes have less interaction with microorganisms because of the membranes’ tendency to form hydrogen bonds with water molecules contacting their surface, developing a thin water film around the membrane. The formed film can inhibit attachment of hydrophobic foulants on the membrane surface [14,15,16]. 

Incorporation of graphene oxide (GO), one of the high-hydrophilic carbonous materials, into the PSF matrix ameliorates membrane fouling resistance. The abundant oxygenated functional groups, including hydroxyl, ketones, carboxyl, and epoxide groups, cohere a large negative zeta potential on the surface of membrane, which can reduce biofouling. In addition to introducing better anti-fouling and antibacterial properties, GO reinforces thermal and mechanical stability as well as the surface area of the PSF membrane [17]. Accordingly, due to the unique properties of GO, it easily combines with polymeric materials and inorganic particles, which can enhance the overall performance of the membrane [18,19]. Polyvinylpyrrolidone (PVP), as a class of hydrophilic polymer, forms uniform pores on the membrane and could improve the antifouling property of PSF/GO membranes [20].

Application of antimicrobial nanomaterials in polymeric matrices, as a recently developed approach to the fouling challenge, has gained considerable attention since they not only address biofouling by improving the hydrophilicity of membranes but also promote the overall proficiency of membranes, i.e., their permeability, selectivity, and mechanical strength. Incorporation of metallic nanoparticles such as zero-valent metals (Ag, Pt, Fe, Zn, Cu) and metal oxides (TiO_2_, ZnO, SiO_2_, ZrO_2_, Al_2_O_3_) is proven to develop the hydrophilicity of polysulfone membranes besides giving morphological, permeability, antifouling characteristics, and mechanical strength to the produced membrane [21,22,23,24]. Although aggregation of nanoparticles, especially in their high concentration, on the polymeric membrane’s surface is problematic and leads to undesirable consequences such as pore blockage, GO-based membranes can overcome this drawback due to the unique properties they own [25]. 

A number of research projects were carried out to develop PSF/GO membranes, but to our knowledge, no research has been carried out to provide a precise investigation of the combination of Pt nanoparticles in NF membranes for water treatment.

In the present research, a novel nanohybrid membrane was fabricated using blends of PSF and GO-infused Pt via the phase inversion method. The use of GO-infused Pt as a surface modifier offers significant improvement in hydrophilicity, which would guarantee high penetration of the permeate and reduction of membrane biofouling via the low interfacial interaction between the surface of the membrane and foulants. Various loading amounts of GO/Pt modifier were studied to investigate their influence on the properties of the membranes, and various analyzing instruments were employed to characterize the prepared membranes in detail. Additionally, the structure of the membrane was simulated using OPEN-MX software, which is based on density functional theory (DFT).

## 2. Materials and Method

### 2.1. Material

In this research, polysulfone p-3500 (MW = 35,000), the polymeric support, and polyvinylpyrrolidone (PVP), the hydrophilic Udel polymers were supplied by Solvay (Bernburg, Germany) and Scharlau (Barcelona, Spain), respectively. The high purity NaCl was from Dr. Mojallali Co. (Tehran, Iran). Natural graphite, platinum salt (H_2_PtCl_6_), sulfuric acid (H_2_SO_4_), hydrogen peroxide (H_2_O_2_), hydrochloric acid (HCl), potassium permanganate (KMnO_4_), and N-methyl pyrollidone were purchased from Sigma-Aldrich. Type 1 agar was purchased from HiMedia (Mumbai, India) and Luria–Bertani (LB) was provided by Merck (Darmstadt, Germany). The nonwoven polyester was obtained from Viledon (Weinheim, Germany).

### 2.2. Synthesis of GO/Pt Nanoparticles

The hummers procedure was used to synthesize GO from natural graphite [26,27]. GO/Pt nanoparticles were prepared according to a previous procedure [28]. First, 0.025 g of H_2_PtCl_6_ was cast in 25 ml ethylene glycol and stirred vigorously for 30 min. Then, 0.04 g of GO was added to the solution under constant stirring for 1 h at ambient temperature. The well-mixed solution was kept under reflux condition using an oil bath at 160 °C for 3 h. The resultant suspension was cooled to ambient temperature in 2 h. Sulfuric acid (0.1 M) was used to adjust the solution’s pH at 3. Then, the solution was shaken for another 24 h. The blended GO/Pt nanoparticles were rinsed several times with water and then filtered to ensure about the particle size distribution. The final sample was desiccated at 160 °C for 1 h in an air oven.

### 2.3. Synthesis of PSF Support Layer

The asymmetric PSF membrane was synthesized according to the common phase inversion method. PSF (1.5 g) and PVP (0.1 g) were added to NMP solvent and blended under constant magnetic stirring at 200 rpm for 12 h at ambient temperature. The prepared sample was left stable for 3–4 h to become free of bubbles, and then was cast on a nonwoven polyester with the casting knife with 150 μm thickness. The resultant film was immediately submerged in a coagulation bath of deionized (DI) water at 25 °C. At the final stage, the fabricated membranes were washed 3 times to remove remaining solvents and non-attached polymers. Then, they were left in DI water for 12 h [29].

### 2.4. Synthesis of Modified PSF Membrane

PSF membranes infused with GO/Pt nanoparticles were synthesized via phase-inversion technique. Various amounts of GO/Pt (0.25, 0.5, 0.75, and 1 wt%) nanoparticles were added to PSF solution. The solution was stirred for 12 h to achieve a homogeneous mixture. Afterward, the prepared mixture was sonicated for 1 h, and subsequently, left overnight for bubble removal. The prepared solution was then cast on nonwoven polyester according to the abovementioned procedure.

### 2.5. Membrane Simulation

Density functional theory or DFT is a famous modeling tool that predicts the structures and molecular properties of materials. DFT provides a relation between theory and experimental study which can identify the geometric, electronic, and spectroscopic properties of the analyzing systems [30]. OPEN-MX is a well-known software package for simulations of nanomaterials based on DFT [31]. In the current work, we employed OPEN-MX as a powerful theoretical software to find structural features of the fabricated membranes.

### 2.6. Characterization

The spectroscopic investigation of the prepared membranes was via the FTIR method using a Bruker-IFS48 (Ettlingen, Germany) in the range of 400–4000 cm^−1^. In order to analyze the hydrophobicity of the neat and GO/Pt nanoparticle-containing PSF membranes in the form of water contact angle, a Canon camera furnished with DorpImage (Tokyo, Japan) was employed. To reduce the experiment errors, all measurements were taken at three random points or three times (for instance, in water flux and nitrate rejection) for each sample and the standard deviation value was applied. Scanning electron microscopy (SEM) was used to investigate the membrane morphology using Philips-X130 apparatus. For this test, the samples were dried in a vacuum stove at 80 °C and then covered with the gold sneeze to improve electrical conductivity. To acquire a cross-sectional surface, the membrane was submerged in the liquid nitrogen ambiance. The frozen sample was broken by a hit so the membrane structure was kept undamaged. The atomic force microscopy (AFM) technique was employed to scrutinize the membrane’s surface roughness using a Nanosurf checking probe-optical microscope lens (EasyScan II, Liestal, Swiss). A sample size of four squares of each of the fabricated membranes (1 cm^2^) was placed onto the glass substrate. The membrane surfaces were scanned within the range of 5 × 5 μm and surface parameters including S_a_, the mean roughness parameter, S_q_, root-mean-squared roughness, and S_y_, the mean difference between the highest peaks and lowest valleys, were characterized. Additionally, inductively coupled plasma mass spectroscopy (AZ instruments, Taichung City, Taiwan) was used to determine the solution concentration [32].

### 2.7. Membranes’ Performance

A dead-end stirred cell, as described in our previous works [33], was applied to calculate the membranes’ water flux and nitrate rejection ability under pressure of 3 bar over 30 cm^2^ effective area of the membranes. The membrane coupons were soaked in deionized water and pre-compressed for 15 min at 3 bar before being used in the main experiments. The permeate water flux calculation was performed by:(1)$$J=\frac{m}{A\Delta t}$$
where *m* (Kg) is the overall permeate weight in *∆t* (min) period on surface of *A* (m^2^) and *J* (L.m^−2^.h^−1^), the effective area of the membrane and pure water flux, respectively. 

The nitrate rejection capability of the samples was calculated as a function of the concentration differences between the feed and permeate solutions, as is given below:(2)$$R\left(\%\right)=\left(1-\frac{C_{p}}{C_{f}}\right)\times 100$$
where *C_p_* and *C_f_*, respectively, are permeate and feed concentrations.

For this purpose, the nitrate concentration of feed was fixed at 50 mgL^−1^ via adding the NaNO_3_ salt to the DI water.

### 2.8. Preparation of Model Bacteria and Antibacterial Performance of the Membrane

*Escherichia coli* (*E. coli*) cells were harvested on LB broth medium in a shaking incubator (Velp-ZX3, Usmate, Italy) under speed of 180 rpm for 10 h at 37 °C, after dilution of the cell suspensions to 10^6^ colony forming units per milliliter or CFU/mL. The resultant suspension was autoclaved (Reihan Teb autoclave, Tehran, Iran) for 20 minutes at 121 °C and then cooled down in a sterilized Petri dish.

With the aim of investigating the bacteriostatic capability of the neat and developed PSF membranes with GO/Pt nanoparticles, the inhibition zone experiment was performed. All the membranes were maintained under ultraviolet radiation for 30 min to kill any unwanted existing bacteria. Membrane sheets were applied on agar plates that consisted of dilute *E. coli* and cultured at 37 °C. After 24 h, the bacterial inhibition zones were recorded for qualitative assessment of the antibacterial behavior of the membranes. 

Moreover, SEM study was performed for quantitative assessment of the bacterial activity of the membrane. In this regard, membranes were covered with *E. coli* suspension (25 mL, 3 × 10^5^ CFU/mL) for 8 h at 37 °C, and the dried samples were analyzed through SEM.

## 3. Results and Discussions

### 3.1. Membrane Model

An OPEN-MX simulator was used to simulate and predict the membrane structure. Figure 1a,b depicts side and top views of the predicted optimum state of Pt nanoparticles’ position on GO, respectively, and Figure 1c shows the optimized structure of nitrate adsorption on a GO/Pt surface.

### 3.2. Characterization

#### 3.2.1. FTIR Test

The chemical bonds on the surface of PSF membranes with embedded GO/Pt (0.75 wt%) nanoparticles were considered via FTIR spectroscopy to confirm deposition of GO/Pt nanoparticle on the PSF matrix. Figure 2 illustrates a broad peak at 3600 cm^−1^ associated with the stretching vibration of (–OH) groups. Peaks around 1581, 1485, and 1105 were attributed to C–C bonds of PSF aromatic rings [34]. The peaks at 1150 and 1250 cm^−1^ were related to the S=O stretching vibration and symmetric C–O–C stretching of PSF, respectively [35]. A 1650 cm^−1^ peak was attributed to the carbonyl stretching vibration of PVP [36]. These functional groups confirm the formation of a hydrophilic membrane due to the insertion of GO/Pt. The possible route of membrane synthesis was through hydrogen bonds between O=S=O polysulfone groups and –COOH groups of GO/Pt, as can be seen in the new peak around 1700 cm^−1^ [37].

#### 3.2.2. SEM Analysis

SEM images of the synthesized samples are presented in Figure 3 to represent more accurately the influence of GO/Pt nanoparticles on the physical properties of the membranes. A general well-known consequence in regards to the morphology of polysulfone membranes is the asymmetric structure of PSF membranes forming in the phase inversion method. The pores’ morphology is affected by the solvent/nonsolvent diffusion rate in the coagulation bath. Upon immersion of the casting solution in the coagulation bath, the interfacial surface is exposed to the coagulant. The exchange rate of solvent and coagulant is so high that the polymer precipitation and solidifying happen almost concurrently, which leads to the generation of a dense skin layer on the top surface [38,39]. On the other hand, the weak affinity of the PSF to water as the nonsolvent results in a shorter time of solvent exchange and a greater exchange rate caused by the larger finger-shape pores [40].

As may be seen from the surface SEM results (Figure 3a,b), insertion of GO/Pt nanoparticles in the PSF polymer matrix brought about a nodular morphology that was clearly observable on the surface of the membrane. The cross-sectional analysis of neat PSF and that infused with GO/Pt (0.75 wt%) (Figure 3c,d) revealed the irregular structure expected according to the above discussion. The images convey that membranes were constructed by three well-known layers: a dense thin skin layer that controlled the solutes’ permeation, a porous finger-like sublayer, and a macro-void structure at the bottom layer that acted as mechanical support. The stabilized GO/Pt nanoparticles on PSF polymer gave rise to the thinner skin layer in the upper layer and larger finger-like pores in the sublayer. These results were related to the combination of highly hydrophilic GO/Pt nanoparticles, which increased the solvent/nonsolvent exchange rate throughout the phase inversion. Generally, higher mutual affinity between the solvent (water) and nonsolvent (GO/Pt) causes a more porous membrane [41,42].

#### 3.2.3. AFM Analysis

AFM analysis was performed for scrutiny of the physiochemical characteristics of PSF and hybrid PSF membrane embedded with GO/Pt (0.75 wt%) nanoparticles (Figure 4). AFM images comprise the light and dark areas which represent the high or ridge positions and low or valley positions of the membrane surfaces, respectively [43]. The figure illustrates the ridge and valley structure in both membranes. The incorporation of GO/Pt nanoparticles in a neat polymeric base resulted in the formation of the rougher surface membrane. This phenomenon is probably related to adding hydrophilic GO/Pt in the casting solution, thereby causing fast diffusion of solvent/nonsolvent within the phase-inversion stage [44]. In addition to this reason, the inequality upsurge in the membrane surface via insertion of hydrophilic GO/Pt nanoparticles could give rise to the formation of the rougher surface [45,46]. Surface roughness characteristics are summarized in Table 1. The rougher surface of the modified membrane containing hydrophilic GO/Pt nanoparticles suggests that it may have had a more highly porous structure than the neat PSF membrane.

#### 3.2.4. Contact Angle Measurement

Another method of judging the hydrophilicity feature of the modified membrane is the measurement of the contact angle (Figure 5).

The modified PSF membrane with GO/Pt nanoparticles demonstrated a lower contact angle than the unmodified one. This is due to the hydrophilic character of GO/Pt, which eases establishing of hydrogen bonds of hydroxyl groups on the membrane surface and water molecules; therefore, water has a better ability to moisten the membrane. Basically, GO/Pt nanoparticles grafted on the PSF polymer surface increase the wettability of the resultant membrane, which can facilitate the biofouling resistance of the membrane [35,47].

#### 3.2.5. Membrane Performance

Membrane permeability has been considered for neat and developed membranes as a function of various GO/Pt nanoparticle loadings (Figure 6). The nitrate solution flux increased with the content of GO/Pt up to 0.75% wt. The probable reason for this behavior was the presence of more hydrophilic sites due to the increasing trend of GO/Pt doping, which can increase the permeability. As was also observed from SEM results, the preparation of a more porous structure, and larger pores, as the GO/Pt hydrophilic nanoparticles increased was responsible for this phenomenon [48]. However, a further increase in GO/Pt (1 wt%) content decreased the nitrate flux. A high amount of hydrophilic additive (GO/Pt) eventuated in a high-viscose casting solution that slows down the solvent/nonsolvent penetration during the phase-inversion process; following this, smaller pores were formed. Moreover, pore blockage by a high content of GO/Pt could be influential. The ruffled position of GO/Pt on the membrane surface reduced the number of accessible hydrophilic functional groups for the permeate.

Another test for evaluation of hydrophilicity by adding GO/Pt nanoparticles in the casting solution was the nitrate rejection ability of the fabricated membranes. It is clear from results shown in Figure 7 that the rejection capacity of the improved membrane improved as the weight percentage of GO/Pt increased up to 0.75, and then decreased as the weight percentage of GO/Pt increased to 1. The increasing trend can be explained by establishing strong bonds of hydrophilic functional groups of GO/Pt and water molecules that deter passage of nitrate molecules. The decreased performance of membranes for nitrate rejection at higher loading of GO/Pt can be explained by the disordered location of GO/Pt on the PSF matrix giving rise to the less hydrophilic functional groups on the surface [49]. As the results convey, the membrane modified with GO/Pt (0.75 wt%) was the highest-yielding membrane.

#### 3.2.6. The Antibacterial Behaviors of Modified and Neat Membranes

In this part, results of antibacterial/anti-biofouling tests are given. In these experiments, *E. coli* was employed as a bacteria sample and its interaction with a neat PSF membrane and hybrid PSF membrane modified with GO/Pt (0.75 wt%) were investigated. The results of the bacteriostatic activity of both membranes are presented in Figure 8. The presence of a high density of *E. coli* cells surrounding the pristine PSF membrane showed less antifouling activity of the membrane. In contrast, the modified membrane revealed bacteria-killing ability due to the low accumulation of *E. coli* cells around the membrane, which was due to the insertion of hydrophilic properties via incorporation of GO/Pt nanoparticles. Four kinds of forces determine the adsorption tendency of microorganism on the membrane surface: electrostatic, hydrogen bonding, hydrophobic, and Van der Waals forces [50].

Biofouling persistence of the membranes was assessed using the SEM method. Attachment of *E. coli* to the membrane surfaces was scanned with an SEM tool that could define the biofouling property of the membranes. The SEM image of the naive PSF membrane (Figure 9a) depicts a high agglomeration of *E. coli* on the surface of the membrane, which shows the biofouling characteristic of the PSF membrane. Nevertheless, the SEM image of the modified membrane (Figure 9b) reveals no dense existence of *E. coli* on the membrane surface that is associated with the hydrophilicity of GO/Pt modifiers. The hydrophilic surface formed a film layer of water on its surface that could prevent bacterial growth and reproduction, resulting in great antibacterial and anti-biofouling results [51,52].

## 4. Conclusions

The present study focused on the nanofiltration of nitrate from aqueous solution in order to achieve fresh water. Polysulfone (PSF) support doped with graphene oxide (GO)/Pt (0, 0.25, 0.5, 0.75, 1 wt%) nanoparticles were developed via the phase inversion technique. The modified membranes were studied with analyzing instruments. FTIR analysis showed the development of a hydrophilic layer of GO/Pt on the membrane surface via the interaction among groups of O=S=O from polysulfone and the –COOH of GO/Pt. The fast solvent/nonsolvent interaction in the inversion phase and generation of a rough surface was confirmed by SEM and AFM analyses. The contact-angle test revealed that the as-prepared membrane consisting of 0.75 wt% GO/Pt nanoparticles was more hydrophilic than the simple membrane. High antibacterial and antifouling properties are the characteristics of the improved membrane. In addition, maximum flux and rejection performance were observed for PSF- GO/Pt (0.75 wt%). The simulation of membrane designs by OPEN-MX simulating software established the optimum deposition of nanoparticles on the PSF membrane’s surface. The findings of this study showed a promising performance of the PSF membranes developed with GO/Pt nanoparticles as modifiers in water treatment applications.

## Figures and Tables

**Figure 1 micromachines-14-00128-f001:**
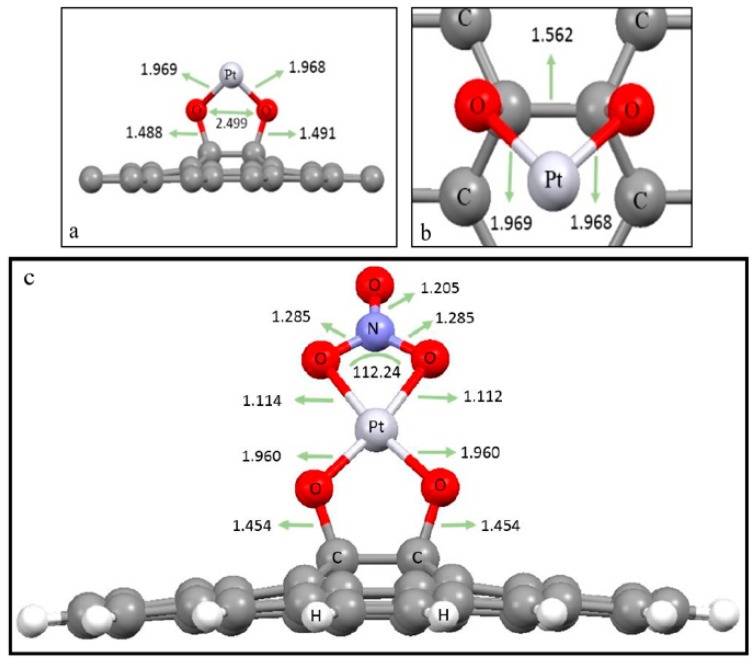
(**a**) Side and (**b**) top views of the optimum position of Pt nanoparticles on GO, and (**c**) the optimized nitrate interaction with the surface of GO/Pt. The bond lengths are represented in Å (angstrom) units.

**Figure 2 micromachines-14-00128-f002:**
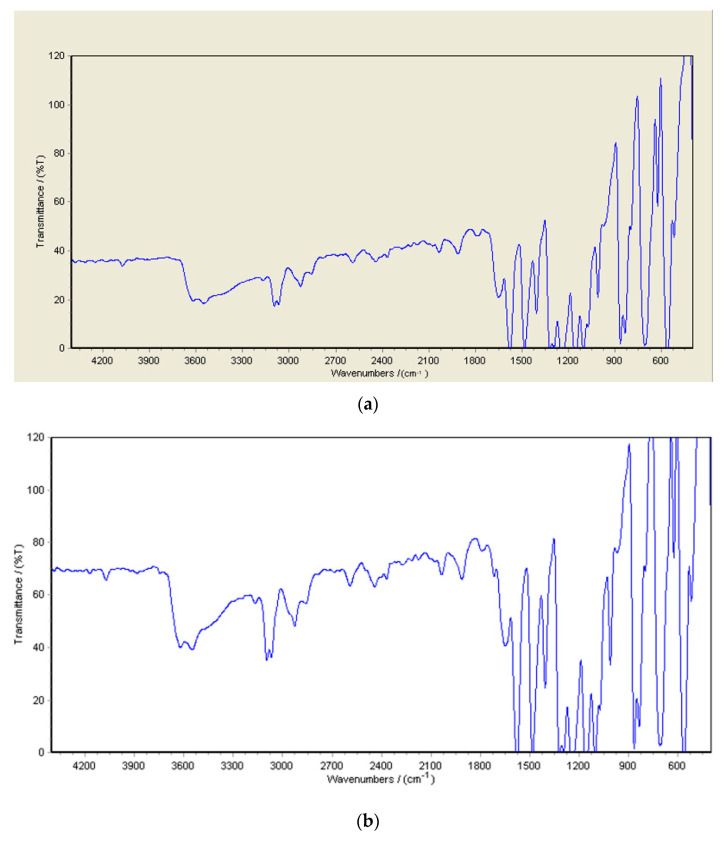
FTIR spectra of (**a**) PSF and (**b**) modified membrane with GO/Pt (0.75 wt%) nanoparticles.

**Figure 3 micromachines-14-00128-f003:**
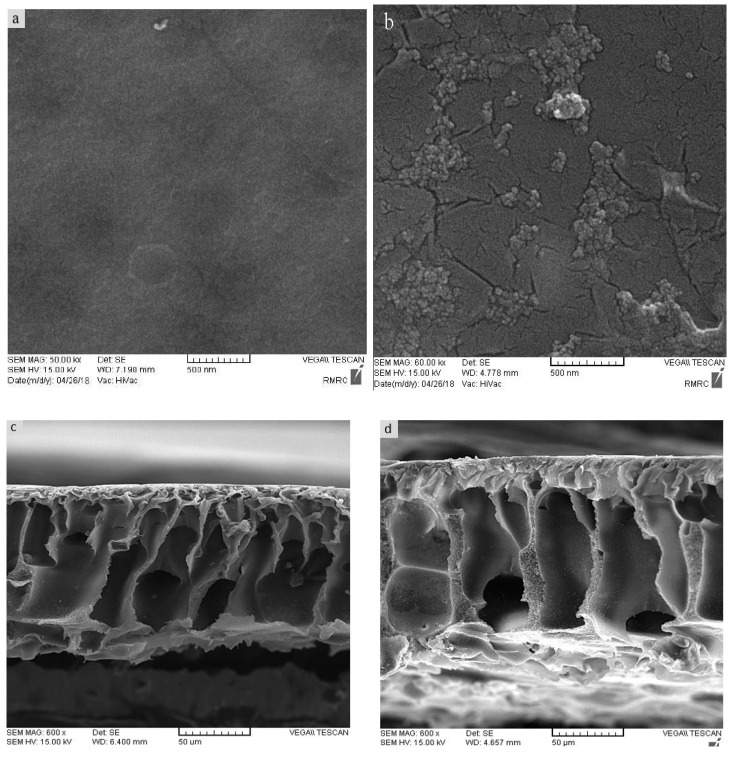
SEM images of prepared membranes’ surface: (**a**) PSF/and (**b**) PSF modified with GO/Pt (0.75 wt%) nanoparticles; cross-sectional SEM images of (**c**) PSF and (**d**) PSF modified with GO/Pt (0.75 wt%) nanoparticles.

**Figure 4 micromachines-14-00128-f004:**
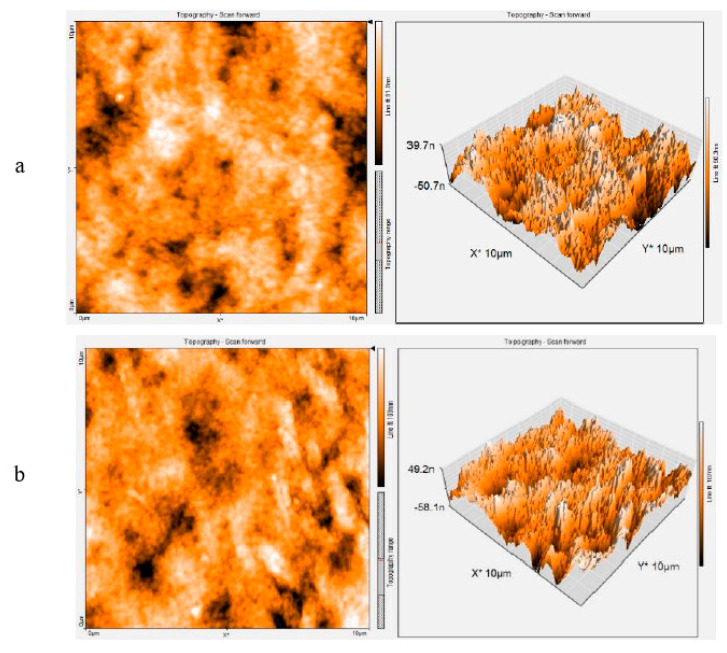
Two-dimensional and three-dimensional AFM images of membranes: (**a**) PSF and (**b**) PSF-GO/Pt (0.75 wt%) nanoparticles.

**Figure 5 micromachines-14-00128-f005:**
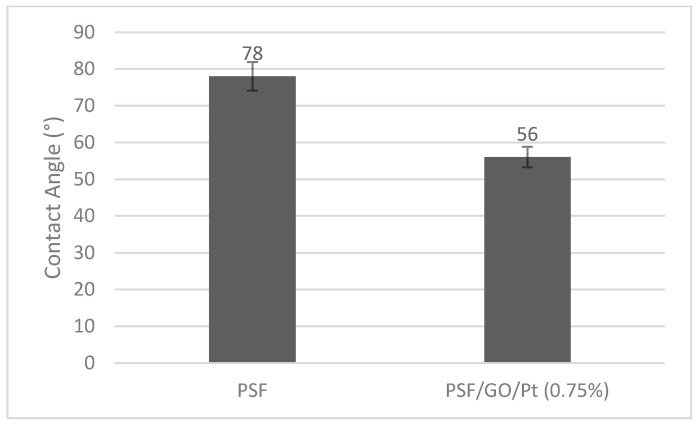
Water contact angle of the neat and modified membranes.

**Figure 6 micromachines-14-00128-f006:**
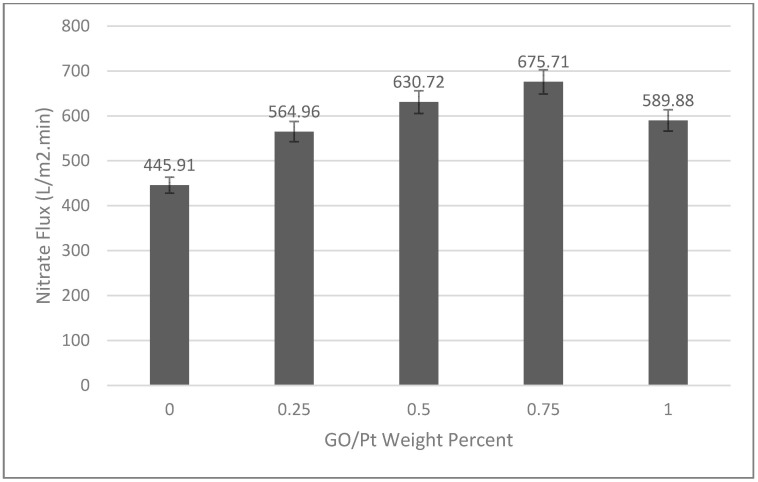
Nitrate flux of prepared polysulfone membranes under different loading percentage of GO/Pt nanoparticles.

**Figure 7 micromachines-14-00128-f007:**
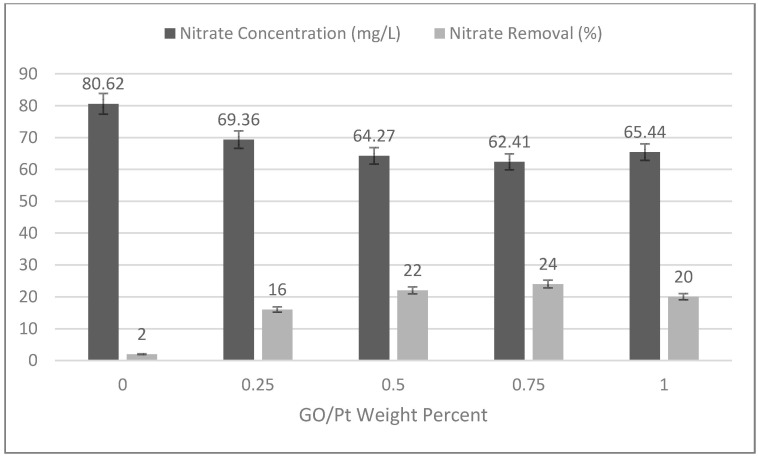
Rejection conditions for nitrate in the polysulfone membranes under different loading percentages of GO/Pt nanoparticles.

**Figure 8 micromachines-14-00128-f008:**
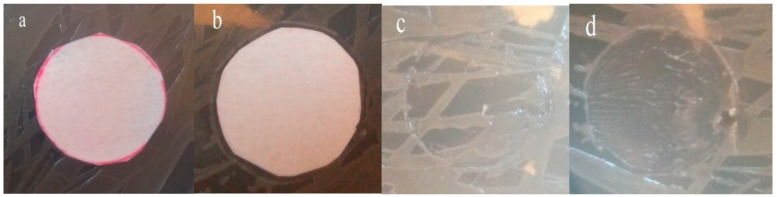
Bacterial growth around (**a**) neat (**b**) and GO/Pt membranes, and (**c**,**d**) under the membranes after a day, respectively.

**Figure 9 micromachines-14-00128-f009:**
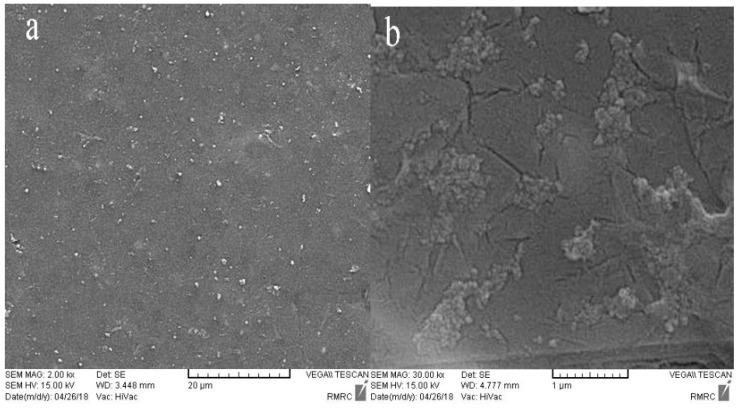
SEM analysis results of E. coli reproduction on (**a**) PSF and (**b**) GO-Pt (0.75 wt%) membranes surfaces.

**Table 1 micromachines-14-00128-t001:** AFM parameters and water contact angles of PSF and PSF-GO/Pt (0.75 wt%) membranes.

Membrane	Sa (nm)	Sq (nm)	Sy (nm)
PSF	16.756	22.218	148.12
PSF/GO/Pt (0.75 %)	21.481	25.863	178.36

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
