# Peer review of "Membrane Surface Modification via In Situ Grafting of GO/Pt Nanoparticles for Nitrate Removal with Anti-Biofouling Properties"

_micromachines, 2023, doi:10.3390/mi14010128_

Round 1

Reviewer 1 Report

Please add error bar for Figure 5 and 6

Please use graph in presenting the contact angle value, and add error bar on it

How many times and on how many samples the water flux and nitrate rejection experiment were done?

Please write the equation using Microsoft Equation, and with uniform font type and font size

How about the effect of GO/Pt on Staphylococcus aureus (Gram-positive bacteria)?

Reviewer 2 Report

The manuscript by Khajouri et al. describes Poly sulfone/PVP nanocomposite with GO/Pt Np membrane system which shows nanofiltration capability in nitrate removal as well as antifouling properties. At an optimum nanocomposite composition, maximum nitrate flux as well as nitrate rejection. Although he material, characterization and performance evaluations look good, I am concerned about clarity of some of the data presented here. The manuscript can be accepted after a  revison considering the following comments.

  1. The manuscript is titled ‘’Anti-biofouling Behavior of in-situ PSF/PVP Composite Coated with GO/Pt Nanoparticles for Nitrate Removal’’ I feel this is inappropriate based on the sequence of results in the manuscript. Is the antifouling behavior necessary for nitrate removal? I feel like these are two different merits of the system. Also, what is the logic of in situ term here? The authors may either explain this or consider an appropriate title change.
  2. The authors are giving quantitation for nitrate flux and nitrate removal. As it is a filtration technique, can you consider nitrate retention in the membrane?
  3. The computational membrane stimulator only considers the GO/Pt part in the structure as shown in Fig 1. What about the molecular interaction of PSF/PVP with GO/Pt nanoparticles? Also, what is the significance of atomic distance values in the computational structure? What is the unit of these values?
  4. The FTIR data should include a comparison data of PSF/PVP neat polymer membrane along with the existing data of nanocomposite to point out the difference
  5. The SEM images show a clear change in the surface morphology of the membrane post-composite formation. The polymer membrane shows an irregular surface on composite formation with GO/Pt which the authors claim to be the reason for the best performance. But the AFM images of neat polymer and their composite counterpart remain the same irregular. How do you explain this?
  6. Figures 5 and 6 should include sufficient error bars to account for the reproducibility of the results.
  7. On page 1, Line 311 it is stated that ‘’The presence of high density of E.coli cells surrounding the pristine PSF membrane shows the less antifouling activity of the membrane. In contrast, the modified membrane revealed bacteria-killing ability due to the low accumulation of E.coli cells around the membrane which is due to the insertion of the hydrophilic property via incorporation of GO/Pt nanoparticles. I don’t see much difference between the two membranes as shown in fFgs 7 C and D. Both of them look the same. I suggest the authors reconsider this experiment with some quantitative data.
  8. Can you consider EDAX measurements on SEM mode to give quantitative evidence of GO/Pt deposition?
